# Comparing the Robustness of Humans and Deep Neural Networks on Facial Expression Recognition

Lucie Lévêque *[ID], François Villoteau [†], Emmanuel V. B. Sampaio [†], Matthieu Perreira Da Silva [ID] and Patrick Le Callet

LS2N—UMR 6004 CNRS, University of Nantes, F-44000 Nantes, France
* Correspondence: lucie.leveque@univ-nantes.fr
† These authors contributed equally to this work.

**Abstract:** Emotion recognition, and more particularly facial expression recognition (FER), has been extensively used for various applications (e.g., human–computer interactions). The ability to automatically recognize facial expressions has been facilitated with recent progress in the fields of computer vision and artificial intelligence. Nonetheless, FER algorithms still seem to face difficulties with image degradations due to real-life conditions (e.g., because of image compression or transmission). In this paper, we propose to investigate the impact of different distortion configurations on a large number of images of faces on human performance, thanks to the conduct of a crowdsourcing experiment. We further compare human performance with two open-source FER algorithms. Results show that, overall, models are more sensitive to distortions than humans—even when fine-tuned. Furthermore, we broach the subject of annotation errors and bias which exist in several well-established datasets, and suggest approaches to improve the latter.

**Keywords:** emotions; facial expression recognition (FER); image quality; robustness; crowdsourcing

## 1. Introduction

Social interactions are based on the ability to accurately detect others' affective states [1]. Emotions therefore play a crucial role in communication, and have been widely studied over the past decades in psychology, but also in the field of computer science—or *affective computing* [2]. Affective computing systems aim, among others, to automatically recognize, and take into account, human emotions during an interaction [3]. This new emotional dimension in computing has proven to be very useful for different applications including, but not limited to, education [4] and mental health [5]. Human emotional states can be estimated using different types of measures, i.e., physiological and non-physiological ones. Physiological measures (e.g., electrocardiogram) are usually invasive. Among the non-physiological ones, facial expressions are particularly interesting as it has been estimated that about 55% of emotional information is conveyed via non-verbal styles of communication [6]. Consequently, automatic facial expression recognition (FER)—made possible thanks to recent advances in computer vision—has emerged as a challenging task, and even more so in natural conditions.

Comparing the performance of humans and machines can help diagnose cases where the latter disagree [7]. In the particular context of computer vision, several works have been proposed to compare computational models with human participants, on diverse classification tasks. Such works generally use a set of stimuli composed of images or videos, further associated to a given label. For instance, Dodge and Karam compared the performance of humans and deep neural network (DNN) models on a classification task [8]. More specifically, their dataset was composed of a large number of images of dogs distorted with Gaussian noise and Gaussian blur at different levels. The authors showed that the performance of humans and machines was approximately the same for

original (i.e., non-distorted) images, while humans' performance was overall more robust to distortions.

Our contributions in this article are the following:

- We introduce Distorted-FER (DisFER), a new facial expression recognition (FER) dataset composed of a wide number of distorted images of faces. Note that our dataset will further be publicly released for the community (it will soon be made publicly available at: ftp://ftp.ivc.polytech.univ-nantes.fr/LS2N_IPI_DisFER, last accessed on 30 October 2022).
- We present a large-scale online experiment conducted on a crowdsourcing platform to assess human performance on a FER task on our dataset.
- We compare the human performance to that of pre-trained and fine-tuned open-source deep neural networks.
- Finally, based on the results obtained, we raise a discussion on how to define labels and on the very definition of what is a facial expression ground truth.

More specifically, the remainder of the paper is structured as follows. Section 2 describes related works. Section 3 deals with the creation of our dataset, the crowdsourcing experiment, and the models applied to our dataset. Section 4 presents obtained results, as well as a comparison between humans' and models' performance. Finally, these results are further analyzed in Section 5, along with a discussion on facial expression labels.

## 2. Related Works

As far as facial expression recognition is concerned, several research works can be found in the recent literature on distorted images of faces or on the comparison of humans' and machines' performance. A few are introduced in this Section.

In 2021, Yang et al. carried out a benchmark on five commercial systems (i.e., Amazon Rekognition, Baidu Research, Face++, Microsoft Azure, and Affectiva) [9]. More precisely, they studied the effects of various manipulations, e.g., rotation, underexposure, and overexposure, on images of faces. The authors showed the limitations of most studied models under image distortions. Furthermore, they recommended the use of different systems depending on the nature of the manipulation (e.g., Baidu better for rotation or low-level noise strength, and Amazon for high-level overexposure or high-level blurring).

More recently, Abate et al. studied how FER solutions deal with faces occluded by facial masks (e.g., worn due to the COVID pandemic) [10]. To do so, they analyzed the performance of three open source deep learning algorithms, i.e., residual masking network, FER-with-CNNs, and amending representation module, on images of masked faces and of eye-occluded faces. They found a significant decrease in recognition accuracy on masked faces, showing that the mouth strongly contributes to expression classification.

Similarly, Poux et al. investigated facial expression recognition in the presence of partial occlusion [11]. They proposed a new method aiming to reconstruct the occluded part of the face for FER. Their method was applied to a wide set of videos of faces, and further compared to four other state-of-the-art approaches. The proposed method showed better results in the context of mouth occlusions and lower part occlusions, where other models faced difficulties.

As for Dupré et al., they analyzed how humans and eight commercially available models (i.e., Affectiva's Affdex, CrowdEmotion's FaceVideo, Emotient's Facet, Microsoft's Cognitive Services, MorphCast's EmotionalTracking, Neurodata Lab's EmotionRecognition, VicarVison's FaceReader, and VisageTechnologies' FaceAnalysis) performed on emotion classification using a large number of video stimuli [12]. According to their results, human subjects reached an average accuracy of 75%, which was higher than the performance of the best model, i.e., Facet, with an accuracy of 62%.

A year later, Krumhuber et al. compared the performance of humans and this model on the recognition of posed and spontaneous expressions taken from different video clips [13]. They showed that both humans and machine tended to perform better on posed expressions,

in accordance with the prior literature. Moreover, they found that Facet performed better than humans on posed datasets—which was not the case for spontaneous expressions.

Earlier this year, Monaro et al. published their work which consists in a comparison between humans and machine learning models on the recognition of facial micro-expressions on a set of videos [14]. The authors made use of different approaches, including machine learning techniques (i.e., support vector machines (SVM) and deep neural networks), and feature extraction methods (i.e., improved dense trajectories and OpenFace). By studying micro-expressions, Monaro et al. aimed to detect liars. Their results showed that models performed better than humans in the lie detection task based on faces.

Yet, to the best of our knowledge, no work has compared the performance of humans and FER algorithms on distorted images of faces, which is what we are going to present in the following sections.

## 3. Materials and Methods

### 3.1. Dataset

The source images used in our experiment come from the Facial Expression Recognition 2013 (FER-2013) dataset [15]. This dataset was firstly introduced in 2013 at the International Conference on Machine Learning, and has been used in a large number of research works since then, as it encompasses naturalistic conditions and challenges. This dataset consists of 35,887 images of faces in $48 \times 48$ format, collected thanks to a Google search. Human accuracy on FER-2013 was estimated by its authors around 65.5% [15].

To build the Distorted-FER (DisFER) dataset, we randomly selected, from FER-2013, twelve images per basic emotion, as defined by Ekman [16] (i.e., anger, disgust, fear, happiness, neutral, sadness, and surprise). This yields a total of 84 source images. Each original stimulus was then distorted using three different types of distortions, i.e., Gaussian blur (GB), Gaussian noise (GN), and salt-and-pepper noise (SP). Each distortion was applied at distinct levels: three standard deviation values were tested for GB, i.e., 0.8, 1.1, and 1.4; similarly for GN with standard deviation values equal to 10, 20, and 30; while probability levels of 0.02, 0.04, and 0.06 were chosen for SP; corresponding to low, medium, and high distortions, respectively. Figure 1 illustrates a sample of distorted images from DisFER dataset, which contains a total of 840 stimuli, including the original.

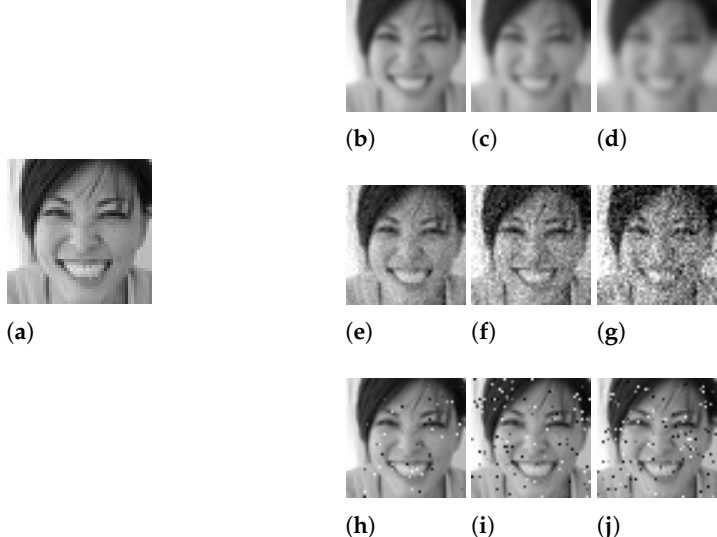

**Figure 1.** Illustration of sample distorted images used in our study: (**a**) represents an original image from the FER-2013 dataset; (**b**–**d**) are its GB versions; (**e**–**g**) the GN versions; and (**h**–**j**) the SP versions (at low, medium, and high levels, respectively).

### 3.2. Crowdsourcing Experiment

In order to collect as many votes as possible on our dataset, and because rating 840 images is time-consuming and can be extremely tiring for a single participant, we decided to set up a crowdsourcing experiment. Such experiments indeed allow the conduct of large-scale subjective tests with reduced costs and efforts [17].

The DisFER dataset was therefore split into twenty-one playlists of forty images each, with a view to keep the tests as fast as possible—as crowdsourcing experiments should not last more than ten minutes or so [18,19]. Playlists were carefully designed to contain the same numbers of images of a given configuration (i.e., emotion, distortion types, and distortion levels). Among a playlist, images were randomly displayed to participants.

Each participant was asked to choose which emotion (i.e., anger, disgust, fear, happiness, neutral, sadness, or surprise) they recognized in the displayed image. No time constraint was imposed on participants to fulfill the task.

A total of 1051 participants (including 50% of females) were recruited using the Prolific platform [20]. Prolific takes into consideration researchers' needs by maintaining a subject recruitment process that is similar to that of a laboratory experiment [21]. Indeed, participants are fully informed that they are being recruited for a research study. Consequently, this platform allows researchers to eliminate ethical concerns, and it further improves the reliability of collected data [22].

Participants were aged between 19 and 75 years old (with a mean of $30 \pm 8.53$—note that three participants did not wish to respond). Twenty playlists out of twenty-one were entirely watched and rated by fifty distinct participants, whereas one playlist was watched and evaluated by fifty-one participants.

### 3.3. Deep Neural Networks

Based on a preliminary benchmark carried out using facial expression recognition (FER) pre-trained models [23], two deep neural network (DNN) architectures were considered in our study, i.e., the residual masking network (RMN) [24], and the convolutional neural network (CNN) offered by DeepFace [25]. Both of these are open source Python libraries. In our preliminary study, RMN achieved an accuracy of 51% on the original FER-2013 dataset, while DeepFace reached 55%.

However, using such pre-trained models suffers from a drawback associated with their poor accuracy on classifying distorted images. According to Dodge and Karam [26], DNN models which were fine-tuned on only one type of distortion do not easily generalize to other distortions. Consequently, both pre-trained models were further fine-tuned for each distortion type (i.e., GB, GN, and SP).

Fine-tuning involved a training process with 27 batches of 42 images. In the training phase, both models used the sparse categorical cross entropy as loss function. Each batch was homogeneously divided among the seven possible emotional classes; half of a batch consisted of distorted images, while the other half was kept undistorted. A total of 756 images for training and 84 images for evaluation were used for each distortion type. Models which performed best during the validation phase were then evaluated using the same 840 DisFER images assessed by human participants in the crowdsourcing experiment.

## 4. Results

### 4.1. Overall Accuracy

A human baseline score was needed in order to be able to contextualize models' performance. Such baseline was computed using the collected votes. More precisely, we defined the human predicted emotion for a given image as the expression selected by the highest number of subjects [12]. Overall, for all images and conditions of the DisFER dataset, this approach yielded a classification accuracy of 63% for human participants when compared to the original FER-2013 labels. It can be noted that this accuracy value is close to the one obtained on the original FER-2013 dataset (i.e., 65.5%) [15].

Similarly, the performance of both pre-trained libraries (i.e., RMN and DeepFace) on DisFER was evaluated using common classification metrics. RMN obtained an accuracy of 26%, while DeepFace reached 33%. This is much lower than their performance on the original dataset (i.e., 51% and 55%, respectively) [23]. Fine-tuning the models increased their accuracy, as both of them achieved 52%. In order to statistically quantify this rise, we performed further data analysis using McNemar's test [27]. More specifically, we made the following $H_0$ assumption: the original model (i.e., RMN or DeepFace) and its fine-tuned version reached the same performance. We obtained *p*-values equal to $10^{-32}$ and $10^{-21}$, respectively, for RMN and DeepFace. These *p*-values are below the 0.05 threshold. Consequently, we can reject the null hypothesis, meaning that there is a significant difference between the performance of RMN and its fine-tuned version, and similarly for DeepFace. It can be noted that there was no significant difference (i.e., *p*-value equal to 0.96) between the fine-tuned versions of RMN and DeepFace.

Figure 2 represents the accuracy values obtained by humans, pre-trained, and fine-tuned models based on the different distortion configurations studied. These values show that humans performed better than machines in terms of facial expression recognition, and this under all distortion types and levels, as well as on original stimuli. Furthermore, humans were more robust to distortions than models, as their accuracy decreased less. More precisely, when making the null hypothesis that humans and the studied model (i.e., RMN, DeepFace, or their fine-tuned version) reached the same performance, we obtained each time (i.e., for the four models) *p*-values close to zero (i.e., $10^{-55}$ for RMN, $10^{-38}$ for DeepFace, $10^{-8}$ for fine-tuned RMN, and $10^{-8}$ for fine-tuned DeepFace). It can also be seen on this figure how fine-tuning both models significantly increased their accuracy, as mentioned in the previous paragraph.

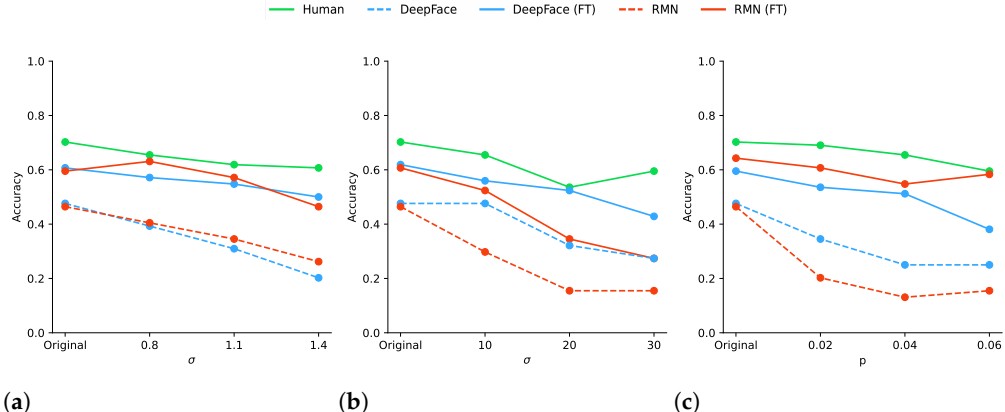

**Figure 2.** Illustration of the accuracy achieved by humans, pre-trained (i.e., DeepFace and RMN) and fine-tuned (FT) models, for each distortion type (i.e., GB, GN, and SP) and level. (**a**) Gaussian blur; (**b**) Gaussian noise; (**c**) salt-and-pepper noise.

In terms of distortions, Gaussian noise seemed to have the greatest impact on both humans and models, as their accuracy decreased the most as levels increased. On the contrary, salt-and-pepper presented different effects on humans and models, as humans were not really affected by distortions, compared to pre-trained models which were strongly impacted. For instance, the accuracy of RMN dropped from 46% to 20% between original images and the first level of SP, while humans' performance remained around 70%.

### 4.2. Confusion Matrices

Figure 3 illustrates the confusion matrices created for humans, pre-trained, and fine-tuned models for each distortion configuration to better understand their impact depending on the studied emotion. Several points regarding the performance of humans and machines can be raised from these matrices.

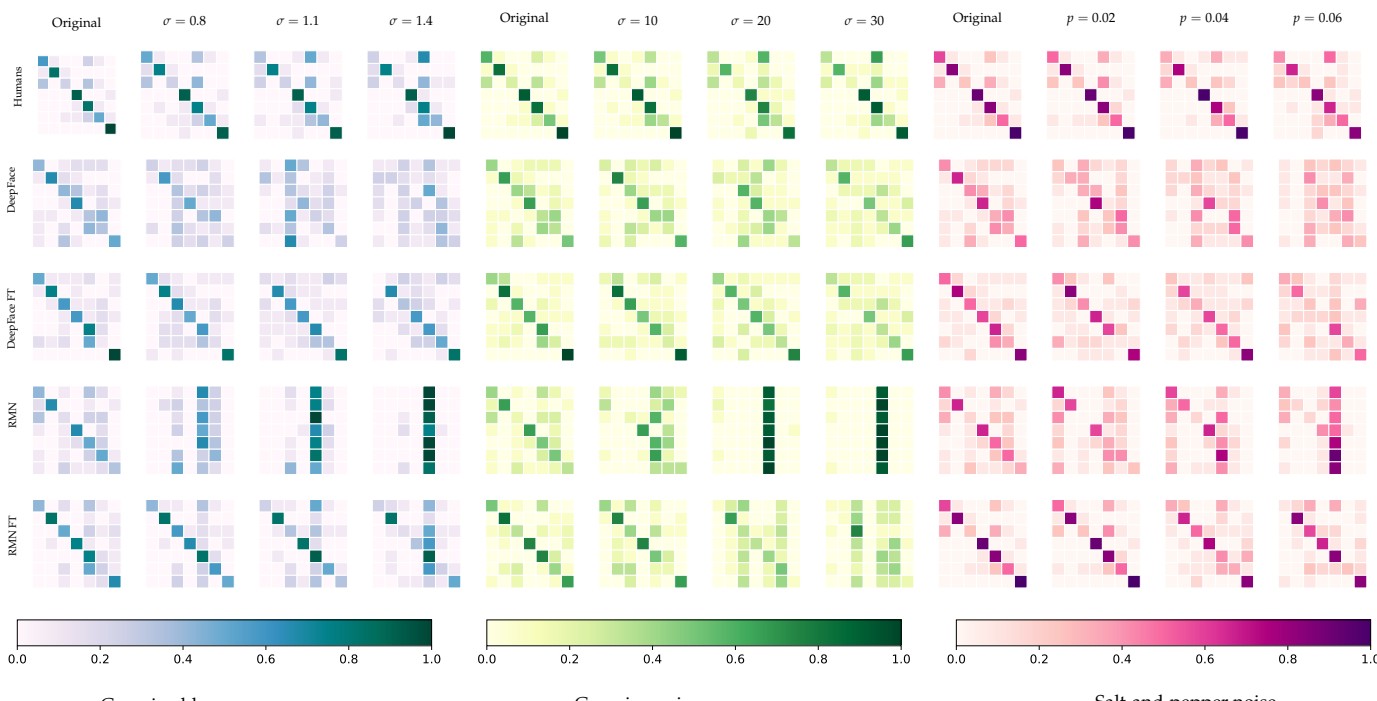

**Figure 3.** Illustration of the confusion matrices generated for humans, pre-trained (i.e., DeepFace and RMN) and fine-tuned (FT) models, for each distortion type (i.e., GB, GN, and SP) and level. Lines correspond to ground truth labels, while columns represent predicted labels. Emotions are presented in alphabetical order, i.e., anger, disgust, fear, happiness, neutral, sadness, and surprise.

Firstly, it is possible to observe a greater homogeneity among humans' votes compared to models, revealed by the matrices' main diagonals—more colored, and this under all studied distortion configurations. Nonetheless, even humans did not achieve homogeneous FER performance, as some emotions have their main diagonal values lower than others. More precisely, surprise was the easiest to recognize, with a precision of 85% and a recall of 94%. It was closely followed by happiness, with a recall of 90%. Yet, some misclassified images bring down its precision to 78%. Neutral was the most voted emotion during the experiment, and therefore presents a fairly low precision (i.e., 38%). Fear obtained the worst recall (i.e., 22%), meaning it was the most difficult to recognize.

Confusion matrices of original and fine-tuned models demonstrate an important difference regarding their ability to classify emotions—fine-tuned models were in general more accurate and more robust. It can be noted that the fine-tuned version of RMN performed slightly better on images distorted with GB (i.e., 55% for RMN vs. 54% for DeepFace), and significantly better with SP (i.e., 58% for RMN vs. 48% for DeepFace); while the fine-tuned version of DeepFace reached a significantly higher performance on images distorted with GN (i.e., 50% for DeepFace vs. 38% for RMN). RMN tended to classify all images as neutral, more particularly for GB and GN, and at intermediate and high distortion levels.

### 4.3. Correlation Between Errors

To further analyze the correlation between humans' and models' errors, we calculated the Pearson's correlation coefficient between their confusion matrices when considering only the non-diagonal elements, as recommended by Borji and Itti [7]. Results are illustrated on Figure 4, which shows that, overall, models present a positive correlation with humans' errors—except for DeepFace under the medium level of GN. However, on average, humans' and models' errors did not show significant degrees of correlation, with Pearson's coefficients equal to 0.50 for RMN, and 0.24 for DeepFace.

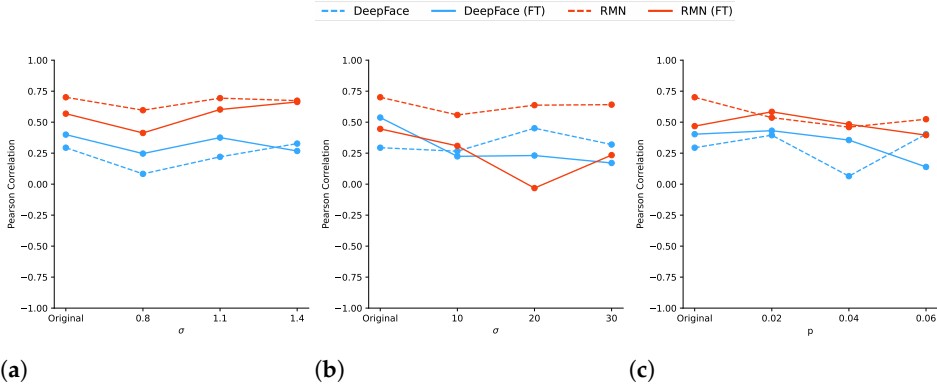

**Figure 4.** Illustration of the Pearson's correlation coefficients calculated between humans' and (original and fine-tuned) models' errors for each distortion type (i.e., GB, GN, and SP) and level. (**a**) Gaussian blur; (**b**) Gaussian noise; (**c**) salt-and-pepper.

It is interesting to notice that, in general, the fine-tuned version of DeepFace obtained higher correlation coefficients than its original version, while the contrary happened for RMN (i.e., errors made by the original version of RMN were closer to humans' errors than errors made by its fine-tuned version). As an example, Figure 5 shows the confusion matrices generated for images distorted using Gaussian blur when removing the main diagonal elements. In this case, we can observe that RMN tended to have prediction errors between the same pairs of emotions for humans, while DeepFace presented more spread errors. This can be explained by the fact that they pay attention to different features in an image to perform their classification.

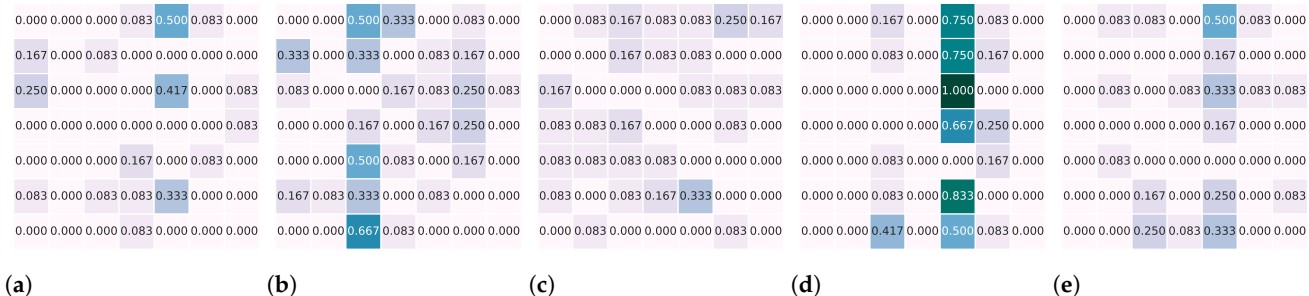

**Figure 5.** Illustration of the confusion matrices generated for humans, pre-trained (i.e., DeepFace and RMN) and fine-tuned (FT) models, for images distorted with Gaussian blur. Main diagonal elements were removed to better compare prediction errors. Lines correspond to ground truth labels, while columns represent predicted labels. Emotions are presented in alphabetical order, i.e., anger, disgust, fear, happiness, neutral, sadness, and surprise. (**a**) Humans; (**b**) DeepFace ($r = 0.33$); (**c**) DeepFace FT ($r = 0.27$); (**d**) RMN ($r = 0.67$); (**e**) RMN FT ($r = 0.66$).

## 5. Discussion

### 5.1. On Obtained Results

We previously showed how humans performed much better than open-source deep neural network (DNN) models when executing facial expression recognition tasks on images with and without distortions. However, even humans tended to perform inconsistently; some emotions caused more disagreement amongst participants than others. Fine-tuning significantly improved the performance of both studied models, yet it was still lower than that of human participants.

According to our observations and to the existing literature (e.g., [9,26]), the human visual system (HVS) is more robust to signal distortions than DNNs. With a view to improve the performance of the latter, not only fine-tuning is needed, with images coming from a context close to the one that will be analyzed; but a better understanding of the

HVS is crucial. Indeed, experience (e.g., prior exposure to distorted stimuli) may play an important role in human performance.

As a matter of fact, over all 840 images and all conditions, when considering the individual performance of each participant (compared to the analysis presented in the previous section), humans obtained an average classification accuracy of 55% $\pm$ 0.4%. The lowest accuracy, i.e., 30% (that is to say 12 correctly classified images), was obtained for three participants; while another one managed to reach a performance of 80% (corresponding to 32 well-classified images).

This brings forward the poor accuracy obtained by human participants on both FER-2013 (65.5%, according to [15]) and DisFER datasets. With a view to analyze this low performance on DisFER, we further compared the original labels with the human prediction baseline for each image of our dataset. More particularly, we put aside the images where all of their ten versions (i.e., original and distorted) were misclassified. A total of 19 contents out of 84 were detected, which corresponds to 23% of the DisFER dataset. When removing these 19 images from the original sources, the mean human accuracy, calculated over all participants over all DisFER images, went up to 82%.

*5.2. On Original Labels*

To illustrate the issue encountered, Figure 6 depicts a few examples taken amongst the 19 cases mentioned above. One can quickly notice that predictions made by the majority of human participants seem to be more accurate than the original labels. Such potential label errors may be due to the fact that the original FER-2013 dataset was generated using the Google image search API to search for images of faces [15].

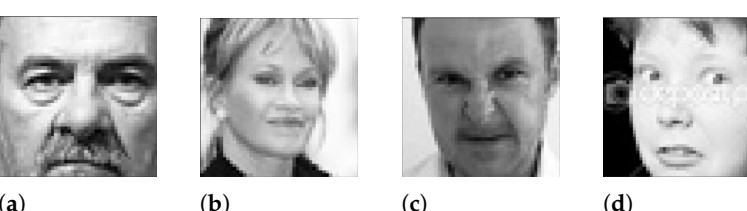

(**a**)          (**b**)          (**c**)          (**d**)

**Figure 6.** Illustration of sample images misclassified by human participants under all of their 10 versions; (**a**) was classified as anger, but labeled as fear; (**b**) classified neutral, labeled sadness; (**c**) classified anger, labeled disgust; and (**d**) classified fear, labeled disgust.

All this raises the following question: what is the definition of ground truth for facial expressions? Usually, ground truth data refer to data collected in real-world settings, or data "known to be true" [28]. However, when talking about emotions and facial expressions, such assumption can be strongly controversial [29].

In order to try to answer this question, we first had a look at the inter-observer agreement, referring to the degree of agreement among several participants who assess a given phenomenon. Let us call $P_i$ the extent to which participants agree for the $i$-th image (i.e., the number of pairs of participants in agreement, relative to the number of all possible pairs). $P_i$ is calculated as follows:

$$P_i = \frac{1}{n(n-1)} \sum_{j=1}^{k} n_{ij}(n_{ij} - 1)$$

where $i$ belongs to [1, 840] (840 being the total number of images), $n$ corresponds to the number of ratings per image (i.e., 50 or 51), and $k$ is the number of categories into which assignments are made (i.e., seven emotions). Consequently, $n_{ij}$ represents the number of participants who assigned the $i$-th image to the $j$-th emotion.

A total of 81 images out of 840 (i.e., 9.6%) of our dataset present a $P_i$ value above 0.9, meaning that participants strongly agreed on these images. One can note that most of these images (i.e., 43 out of 81) were classified in the happiness category, which seems to be the

easiest one. A total of 513 images (i.e., 61%) obtain a $P_i$ value above 0.5, corresponding to a moderate agreement among participants. Conversely, 88 images (10%) are linked to a $P_i$ below 0.3, showing a very poor degree of agreement.

With a view to come back to our aforementioned issue, we then investigated the $P_i$ values calculated for the nineteen contents which were misclassified under their ten versions (i.e., original and distorted) by most participants. Table 1 represents the $P_i$ values obtained for the four samples illustrated in Figure 6. Obtained values show a fair agreement among observers for images a and d, and a moderate agreement for images b and c.

**Table 1.** Participants' degree of agreement $P_i$ calculated for a sample of images $i$ misclassified under their ten presented versions. Note the images are the ones illustrated on Figure 6.

| $P_i$ | Figure 6a | Figure 6b | Figure 6c | Figure 6d |
|---|---|---|---|---|
| Mean | 0.34 | 0.41 | 0.54 | 0.38 |
| Standard dev. | 0.11 | 0.13 | 0.09 | 0.07 |
| Minimum | 0.30 | 0.30 | 0.42 | 0.27 |
| Maximum | 0.43 | 0.49 | 0.70 | 0.53 |

When considering the specific example of image b, labeled by FER-2013 dataset under the sadness category, it only received 2% of votes for this emotion. However, it was classified as happiness by 42% of participants, and as neutral by 47%. Similarly, when taking the case of image c, originally labeled as disgust, 22% of human predictions correspond to this emotion, whereas 54% are linked to anger.

To assess the overall reliability of agreement over all raters and all images of the DisFER dataset, we made use of Fleiss' kappa ($\kappa$) [30], which is calculated as follows:

$$\kappa = \frac{\bar{P} - \bar{P}_e}{1 - \bar{P}_e}$$

where $\bar{P}$ is the mean of the $P_i$'s over all $N = 840$ images, i.e.,

$$\bar{P} = \frac{1}{N} \sum_{i=1}^{N} P_i$$

and $\bar{P}_e$ is calculated as follows:

$$\bar{P}_e = \sum_{j=1}^{k} p_j^2$$

where $p_j$ is the proportion of classifications made for the $j$-th category (the sum of $p_j$'s over the seven basic emotions being equal to 1).

More precisely, the factor $1 - \bar{P}_e$ corresponds to the degree of agreement reachable above chance, and $\bar{P} - \bar{P}_e$ corresponds to the degree of agreement actually achieved above chance. When raters are in complete agreement, $\kappa = 1$. On the contrary, if there is no agreement at all, then $\kappa \leq 0$. Human participants reached a Fleiss' kappa equal to 0.58 on our dataset, which represents a moderate agreement [31]. This further shows that recognizing emotions is not straightforward, even for humans.

### 5.3. On a New Ground Truth Definition

These results can put some of the original labels under question. Indeed, in cases where participants strongly agree on an emotion, but disagree with the original label, an incorrect ground truth annotation would explain the low accuracy values obtained. Furthermore, in cases where participants' votes are more or less equally split between two or more emotions, problems may arise if labels are defined by a single annotator, as for the AffectNet dataset [32].

Consequently, we would first recommend to define ground truth labels based on human participants' classifications rather than on search engines' labels. Moreover, when images achieve a high degree of agreement, labels can be defined using only one emotion—as currently done in most existing FER datasets—but it could be interesting to have two (or more) labels when humans tend to disagree. Such low data reliability yields crucial challenges for FER technologies [33]. In this context, FER models could learn from several emotions, as well as a human percentage of confidence, which would help improve their performance.

## 6. Conclusions

In this article, we presented a study to compare the performance of human participants and deep neural networks on facial expression recognition (FER) under various distortion conditions (i.e., different types and levels). Data analysis indicated that humans performed better and were more robust to image distortions than open source models, and even than their fine-tuned versions.

Yet, we also showed that some of the original labels appeared as dubious when compared to the classifications made by a majority of human participants. Therefore, we opened a discussion on the very definition of ground truth in the context of facial expressions, and suggested new ways to define such ground truth.

In practice, our work opens several questions regarding the robustness of deep FER models. In this view, performing data augmentation by including distorted images when training deep learning models for FER seems to be a sound practice. However, this approach is clearly not sufficient, and further work is required to better deal with this issue.

Moreover, we would argue that these models could, and perhaps should, be trained as regression models instead of classification, by comparing the output probability vector to a ground-truth vector built using the annotations of several observers. This way, the diversity of human opinion and perception would be taken into account, and borderline cases could also be dealt with more accuracy.

**Author Contributions:** Conceptualization, L.L. and M.P.D.S.; methodology, F.V.; software, E.V.B.S.; validation, E.V.B.S.; formal analysis, F.V.; writing—original draft preparation, L.L.; writing—review and editing, L.L. and M.P.D.S.; supervision, M.P.D.S.; funding acquisition, M.P.D.S. and P.L.C. All authors have read and agreed to the published version of the manuscript.

**Funding:** This research received no external funding.

**Informed Consent Statement:** Informed consent was obtained from all subjects involved in the study.

**Data Availability Statement:** The DisFER dataset will be made publicly available at: ftp://ftp.ivc.polytech.univ-nantes.fr/LS2N_IPI_DisFER (last accessed on 30 October 2022).

**Conflicts of Interest:** The authors declare no conflict of interest.

## Abbreviations

The following abbreviations are used in this manuscript:

| | |
|---|---|
| CNN | Convolutional Neural Network |
| DNN | Deep Neural Network |
| FER | Facial Expression Recognition |
| FT | Fine-Tuned |
| GB | Gaussian Blur |
| GN | Gaussian Noise |
| RMN | Residual Masking Network |
| SP | Salt-And-Pepper |

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
