# Peer review of "Comparing the Robustness of Humans and Deep Neural Networks on Facial Expression Recognition"

_electronics, doi:10.3390/electronics11234030_

Round 1

Reviewer 1 Report

The paper deal with a comparison of Facial Expression Recognition between humans and Deep Learning Networks algorithms based on a new dataset the authors introduced called DisFER. 

Generally, the paper is well structured and written, nonetheless, it lacks innovation and the results appear trivial and not interesting.  

Another important issue concerns statistical analysis. In fact, in the title and keywords, the authors use the term "robustness" but there is no sign of any robust statistics inside the paper, despite a strong robust statistical analysis would have been necessary and enhanced the appeal of the work. For instance, the authors state conclusions such as "human were more robust to distortions than models, as their accuracy decreases less" (page 4) without any statistical measure that supports that statement.

The online experiment for labeling the DisFER images by humans is definitely worthy, but then the results obtained comparing humans' performance to Deep Learning Networks algorithms do not provide any contribution to the literature.

Reviewer 2 Report

Dear authors,

I would like to thank you for your efforts writing this paper. While the main contributions of your work have merits, I believe that there are areas for revisions as follows:

-       At the end of the introduction, add a statement to describe the structure of the paper.

-       Page 2, line 53 “DisFER”, define in full in its first appearance.

-       I recommend the suthors to add a section of “Related work” that thoroughly and succinctly review the current state of the literature. More recent references should be cited.

-       Page 3, line 92, “the Prolific platform”, explain further the subject recruitment process.

-       Page 5, line 166, “except for DeepFace under the medium level of GN”, is there a rationale for this outcome? Was this the case in previous research?

-       The conclusion section needs further enhancement. For instance, (1) What are the practical implications of your work? (2) Describe the way forward, e.g., How can future work build upon your research outcomes? All these issues can be discussed in a very succinct manner and can enhance the overall quality of this manuscript.  

I wish you all the best with your manuscript

Round 2

Reviewer 1 Report

I appreciated the great effort made by the authors to improve the quality of the manuscript. The modifications, especially those of the conclusion section, allow the reader to correctly frame the work. With these changes the manuscript reaches a sufficient level for publication.